# An Update of G-Protein-Coupled Receptor Signaling and Its Deregulation in Gastric Carcinogenesis

**DOI:** 10.3390/cancers15030736

**Published:** 2023-01-25

**Authors:** Huan Yan, Jing-Ling Zhang, Kam-Tong Leung, Kwok-Wai Lo, Jun Yu, Ka-Fai To, Wei Kang

**Affiliations:** 1Department of Anatomical and Cellular Pathology, State Key Laboratory of Translational Oncology, Sir Y.K. Pao Cancer Center, Prince of Wales Hospital, The Chinese University of Hong Kong, Hong Kong 999077, China; 2State Key Laboratory of Digestive Disease, Institute of Digestive Disease, The Chinese University of Hong Kong, Hong Kong 999077, China; 3CUHK-Shenzhen Research Institute, The Chinese University of Hong Kong, Shenzhen 518000, China; 4Department of Pediatrics, The Chinese University of Hong Kong, Hong Kong 999077, China; 5Department of Medicine and Therapeutics, The Chinese University of Hong Kong, Hong Kong 999077, China

**Keywords:** G-protein-coupled receptor, G protein, gastric cancer, targeted therapy

## Abstract

**Simple Summary:**

Gastric cancer (GC) ranks as one of the most life-threatening malignancies worldwide, and over one billion new cases and 783,000 deaths were reported last year. The incidence of GC is exceptionally high in Asian countries. Multiple oncogenic signaling pathways are aberrantly activated and implicated in gastric carcinogenesis, leading to the malignant phenotype acquisition. G-protein-coupled receptor (GPCR) signaling is one of them, and the aberrant activation of GPCRs and G proteins promotes GC progression. The activated GPCRs/G proteins might serve as useful biomarkers for early diagnosis, prognostic prediction, and even clinically therapeutic targets. This review summarized the recent research progress of GPCRs and highlighted their mechanisms in tumorigenesis, especially in GC initiation and progression.

**Abstract:**

G-protein-coupled receptors (GPCRs) belong to a cell surface receptor superfamily responding to a wide range of external signals. The binding of extracellular ligands to GPCRs activates a heterotrimeric G protein and triggers the production of numerous secondary messengers, which transduce the extracellular signals into cellular responses. GPCR signaling is crucial and imperative for maintaining normal tissue homeostasis. High-throughput sequencing analyses revealed the occurrence of the genetic aberrations of GPCRs and G proteins in multiple malignancies. The altered GPCRs/G proteins serve as valuable biomarkers for early diagnosis, prognostic prediction, and pharmacological targets. Furthermore, the dysregulation of GPCR signaling contributes to tumor initiation and development. In this review, we have summarized the research progress of GPCRs and highlighted their mechanisms in gastric cancer (GC). The aberrant activation of GPCRs promotes GC cell proliferation and metastasis, remodels the tumor microenvironment, and boosts immune escape. Through deep investigation, novel therapeutic strategies for targeting GPCR activation have been developed, and the final aim is to eliminate GPCR-driven gastric carcinogenesis.

## 1. Introduction

Gastric cancer (GC) is a substantial global health burden, accounting for the fifth most commonly diagnosed cancer and the third leading cause of fatal malignancies worldwide. Incidence rates are markedly increased in Eastern Asia, especially in Mongolia, Japan, and Korea, which are strongly associated with various predisposing and etiological factors, according to several migrant studies [1,2]. Most GC-related deaths occur due to late diagnosis, lymph node metastasis, and refractory after surgery. Thus, numerous efforts have been made to develop useful prognosis markers for early detection and therapeutic targets to improve clinical outcomes. Heterogeneity represents one of the biggest challenges in GC treatment owing to the histological categories and diverse molecular drivers. The well-established histological classification divides gastric carcinomas into diffuse and intestinal types [3]. The Cancer Genome Atlas (TCGA) network also reaffirmed our understanding of molecular categories by analyzing the dysregulated pathways identified in multiomics data. This study developed a robust molecular classification scheme comprising Epstein-Barr virus (EBV), microsatellite instability (MSI), chromosomal instability (CIN), and genomically stable (GS) tumors [4]. In the past two decades, trastuzumab and chemotherapy were used as the first-line treatment, and the combination of ramucirumab and paclitaxel was used in second-line treatment [5]. However, the clinical applicability remains quite limited. There is an urgent need to uncover more targetable pathways to develop more accurate diagnosis makers against nonspecific symptoms in early-stage GC and optimize existing therapy for precision medicine.

Since G protein-coupled receptors (GPCRs) were reported in cellular transformation in 1986, emerging evidence shows that these membrane-embedded receptors regulate many biological processes and are crucial targets against several human malignancies [6]. The involvement of GPCRs in GC is emerging due to the identification of genomic aberrations that lurk at different stages and subtypes of GC and promote tumor initiation and progression [7]. This review recapitulated the current knowledge related to the aberrated regulation of the GPCR pathway in GC, including the common tactic hijacked by tumor cells for their growth, metastasis, and immune evasion. Moreover, we will discuss the advances in the current treatment strategies and summarize the ongoing clinical trials that attempt to translate biological findings into clinical applications.

## 2. Basic Knowledge of GPCRs

GPCRs comprise over 800 members accounting for about 4% of human genes. They have various structures and signal transduction. Based on their specific characteristics, GPCR members are further classified into different subgroups and participate in various physiological processes, whereas the aberrant expression and abnormal activation of GPCRs are associated with tumor progression.

### 2.1. Structure and Classification of GPCRs

GPCRs have seven transmembrane α-helices (TM1-7) that connect the N-terminal extracellular domain (ECD), three extracellular and intracellular loops: ECL1-3, ICL1-3, and the C-terminus (Figure 1). They are classified into six groups based on their structural and functional similarities, whereas only four groups (A, B, C, and F) are found in vertebrates. The Rhodopsin-like class A, which has 719 members, represents humans’ most common but diverse group. Half of the class A members serve as sensor receptors primarily in smell and vision. In contrast, diffusible ligands, such as peptides, lipids, hormones, and nucleotides, can trigger the other receptors. Class B includes secretin and adhesion receptors, which have a similar sequence in 7TM but different sequences in ECD. The secretin subgroup contains receptors for polypeptide gut hormones, such as the GLP-1 receptor, glucagon receptor, and parathyroid hormone receptor. Research has focused on the adhesion receptors by defining the mechanisms of ligand binding sites and the GPCR autoproteolysis-inducing domain (GAIN)-mediated receptor activation. The metabotropic glutamate family (class C) is characterized by a large ECD, consisting of γ-aminobutyric acid B receptors (GABA_B_), metabotropic glutamate receptors (mGluRs), and a calcium-sensing receptor (CasR). The frizzled/taste family (class F) includes frizzled and smoothened proteins that can be activated by the lipo-glycoproteins of the Wnt and Hedgehog families [8,9,10].

Thrilling technologies, such as X-ray crystallography, cryo-electron microscopy (cryo-EM) [11,12], and nuclear magnetic resonance (NMR) spectroscopy [13], are facilitating the exploration of the structures of GPCRs and further boosting the structure-based drug design. More extensive knowledge of GPCR biology, particularly on the function-related conformational equilibria, such as allosteric coupling, biased signaling, and dynamic modulation, is required for developing new targets and minimizing the side effects. Although structural studies reveal the direct impacts of stimulation on the receptor conformation changes, the result of the GPCR signaling pathway is regulated by various factors, including cell backgrounds, receptor expression levels, and agonists’ kinetic characteristics.

### 2.2. Signal Transduction of GPCRs

By binding to various extracellular ligands, GPCRs change their conformation, activate G-coupling proteins, and couple with other proteins, such as β-arrestins and GPCR kinases (GRKs). However, the determinants of GPCR selective binding have not been fully understood. Fundamentally, the G-protein barcode determines the selectivity between GPCRs and G proteins, which contain variable residues on the conserved positions and can be recognized by different GPCRs [14]. G proteins have long been recognized as the primary transducers of GPCRs, and β-arrestins are suggested as essential modulators in genome-edited cells [15]. β-arrestins can identify and bind to the GRK-phosphorylated GPCRs and thereby outcompete the G proteins, functioning as scaffolds for other signalings [16,17]. Therefore, the GRKs regulate the phosphorylation of ligand-bound GPCRs and balance the G-protein-dependent and β-arrestin-dependent nodes of GPCRs [18].

When the activated GPCRs bind with the heterotrimeric G proteins, the Gα subunits dissociate from Gβγ after the exchange of GDP with GTP on the Gα proteins. This process releases the heteromeric G proteins from the GPCRs and retains the plasma’s GTP-bound Gα and Gβγ subunits [19]. One GPCR can activate multiple Gα proteins encoded by 16 genes. These genes are classified into four subfamilies based on sequence similarity: Gi, Gq, Gs, and G12/13. Generally, the Gαs and Gαi respectively promote or inhibit adenylyl cyclase, thus intervening in cyclic adenosine monophosphate (cAMP) production. The accumulating cAMP behaves as a second messenger to activate protein kinase A (PKA). Members of the Gi family also activate phospholipase (PIs) and phosphodiesterases (PDEs), ultimately modulating the opening of numerous ion channels. PLC-β is the effector of the Gαq and Gβγ subunits that elevate intracellular Ca^2+^ levels and activate protein kinase C (PKC) by converting phosphatidylinositol 4,5-biphosphate (PIP2) into inositol 1,4,5-triphosphate (PIP3) and diacylglycerol (DAG). Additionally, the Gβγ subunits trigger the PI3K-γ/PI3K-β to catalyze the conversion of PIP2 to PIP3, which is a direct stimulator of AKT. The Gα12/13 and Gαq family members regulate a group of Rho GEFs, which harbor an RGS homology domain and activate Rho GTPase. As shown in Figure 2A, the segregated Gα and Gβγ subunits evoke kinase cascades by activating various second messengers [20,21].

β-arrestins undergo conformational changes when recognizing the GRK-phosphorylated GPCRs. Then, they enhance the process of desensitization, internalization, and clathrin-mediated endocytosis of the activated GPCRs. As scaffold proteins, β-arrestins facilitate GPCR-stimulated signal transduction. As one of the most prominent and earliest examples, the GPCR-mediated extracellular signal-regulated kinase 1/2 (ERK1/2) activation is a β-arrestin-dependent and G protein-independent signaling [16]. The genetic ablation or inactivation of several G proteins induces a zero functional state for the G protein and abolishes the β-arrestin-mediated signaling in response to GPCR activation [22]. However, it was reported that β-arrestins are not required for ERK1/2 phosphorylation despite their crucial roles in receptor internalization [23]. Indeed, the cumulative impact of GPCR-induced ERK1/2 activation is tightly controlled by β-arrestins and G proteins [24]. Moreover, GPCRs scaffold several signaling proteins for Wnt [25], the hedgehog (Hh) [26], and Notch [27] pathways (Figure 2B,C).

### 2.3. Diversification of GPCR Machinery

GPCRs are sophisticated dynamic machines rather than static on-and-off switches. When they are engaged with different ligands, receptors, and regulatory partners, they may exhibit specific conformations and undergo subcellular distributions. Exploring the dynamic nature of GPCRs is vital to elucidate the mechanisms underlying allosteric modulation, biased agonism, oligomerization, and sustained and compartmentalized signaling. These mechanisms convey novel insights into drug discovery.

Allosteric ligand binding sites in GPCRs are potential new targets for modulating GPCR functions and improving drug selectivity. These modulators augment (positive allosteric modulators [PAMs]) or reduce (negative allosteric modulators [NAMs]) the affinity and efficacy of endogenous agonists [28]. The discovery of allosteric modulators has sparked interest in central nervous system (CNS) diseases, though with limited success [29]. MK-7622 is a PAM, selectively binding with the M_1_ muscarinic receptor in the CNS, which has been stopped because it fails to improve recognition and increases adverse effects like diarrhea [30,31].

Another ligand-receptor dynamic is biased agonism, a mechanism in which the active conformational states of the receptors are stabilized by some ligands, resulting in distinct cellular signaling profiles [32]. There are three different modes of biased signaling, including the same receptor bound with other ligands adopting distinct conformations (ligand bias), varying stoichiometric ratios of signaling effectors in distinguished cells (system bias), and GPCR stimulation within divergent intracellular compartments (location bias) [17]. However, substantial evidence on this is limited. The endogenous ligands, CCL9 and CCL21, have been considered equipotent for activating CCR7-G protein coupling and calcium mobilization. However, both ligands cause the distinct conformation of CCR7. Only CCL9 can promote robust receptor desensitization after coupling the β-arrestins and efficiently accelerating ERK1/2 phosphorylation, which CCL21 cannot achieve [33]. Besides, small molecules targeting TRV130 and PZM21 have been utilized to improve analgesia with fewer side effects because of the biased receptor μ-OR activity, potent Gαi signaling profile, and limited β-arrestin recruitment [34,35]. These two examples can partly explain how ligands trigger the biased mechanism of GPCRs. Revealing the structural features of GPCRs under multiple activation states and different cellular backgrounds may be required to understand the biased signaling.

Receptor oligomerization conveys much more diversities in the function and physiological roles of GPCRs. However, unlike the oligomer tyrosine kinase receptors and ion channels, the formation of GPCR multimers remains controversial [36,37]. It has been found that dimerization was necessary for some GPCRs, such as the class C members. The first tangible evidence for GPCR dimerization was that the gonadotrophin-releasing hormone (GnRH) antagonist-conjugated bivalent antibody played an essential role in biphasic receptor formation [38,39]. The emergence of heteromers was associated with the preferential pattern of receptors in different tissues and cell types [40]. The balance between the monomers and heteromers of GPCRs may contribute to diseases [41,42]. Unraveling the pattern of GPCR heteromers will provide pharmacotherapeutic targets to benefit disease management.

Compartmentalized signaling may partly explain why the GPCRs can activate a typical profile of secondary messengers and kinases. In addition to locating the membrane, the GPCRs might be desensitized and undergo β-arrestin-mediated endocytosis and intracellular signaling [43,44,45]. Notably, some of the mechanisms are studied in the digestive systems. Recently, PAR2 endosomal may underlie the sustained hyperexcitability of nociceptors in patients with irritable bowel syndrome (IBS). The IBS supernatants and trypsin could persistently activate PAR2 in the colonic mucosa in a clathrin-mediated, endocytosis-dependent fashion [46,47]. The inhibitors of clathrin-mediated endocytosis and targeted PAR2 antagonists suppressed PAR2 endosomal signal [48].

### 2.4. Dysregulated GPCR Signaling in Tumors

Based on the recent pan-cancer analysis, GPCR signaling was among the 55 pathways most significantly mutated. Mutations and the aberrant expression of GPCRs and G proteins contribute to various diseases, including neurodegenerative, reproductive, immunological, and metabolic disorders, as well as cancers and infectious diseases [49,50]. The dysregulated GPCR signaling may exert a significant tumorigenic effect, as those alterations frequently co-occur in well-characterized oncogenes, such as tyrosine and serine-threonine kinase Ras-family members [51]. In-depth omics analysis approaches, like MutSig2CV and GISTIC (Genomic Identification of Significant Targets in Cancers), have comprehensively investigated the mutations and copy number variations (CNVs) of GPCRs and G proteins in 33 TCGA patient groups. Remarkably, mutated GPCRs and G proteins have been significantly identified in GI malignancies, even though these tumors’ mutation rates are not typically high [52]. Therefore, the relevance between these mutations and biological outcomes is vastly underestimated [53,54].

### 2.5. GPCR Mutation and Abundant Expression

GPCRs are mutated in over 20% of all sequenced samples [55,56]. Unlike the mutated hotspots in G proteins, GPCRs exhibit diverse mutations across different cancer types. The three-dimensional structures of GPCRs and their interaction elements were evaluated to acquire a mutational landscape of GPCRs in cancers [51]. The bulk of the alterations occurs in the conserved 7TM via the visualization of the representative GPCR 3D structure, such as the ionic lock switch E/DRY arginine motif, G protein-binding sites, and the tyrosine toggle switch motif NPxxY, and ligand-binding site. GPCR mutations impair GPCR signaling by altering the basal activity, ligand binding affinity, G-protein interaction, and cell-surface expression. As with thyroid-stimulating hormone receptors like *HCRT2*, *P2RY12*, *LPAR4*, and *GPR174*, frequent mutations in the DRY motif may result in constitutive activation due to conformational changes in TM3, TM5, and TM6 [51]. Mutations of the connection between the NPxxY motif on TM7 and a conserved tyrosine in TM5 could stabilize the inactive-state conformations of the α_1B_- and β_2_-adrenoceptors, which may account for lower agonist potency in transducing the downstream IP1 and cAMP signaling, respectively [57]. Understanding the mutated structural features will shed new light on GPCR malfunctions and devise possible therapeutic strategies [58].

The large and ever-grossing body of sequencing by pan-cancer analysis suggests that the frequently mutated GPCR families are adhesion-related GPCRs, such as the glutamate metabotropic receptors (*GRM1-8*, class C) [59], lysophosphatidic acid (LPA) receptors (*LPAR1-6*), sphingosine-1-phosphate (S1P) receptors (*S1PR1-5*), and muscarinic receptors (*CHRM1-5*, class A). However, most adhesion receptors are orphan receptors with unknown ligands and physiological functions [60,61]. GPCR genetic alternations were found in melanoma by exon capture and massively parallel sequencing. *GPR98* and *GRM3* were two of the most frequently mutated genes, with 27.5% and 16.3% mutation rates. GRM3 mutants selectively mediate the MEK signaling that contributes to tumor growth in melanoma, acting as an indicator for patient stratification and precision medicine [62]. MutSig2CV analysis suggests the three most mutant GPCRs in colon cancer are *GPR98* (21.25%), *TSHR* (13.90%), and *BAI3* (13.62%), while *CELSR1* (11.20%), *EDNRB* (8.14%), and *GPR45* (5.09%) account for the three most frequently mutated GPCRs in GC. However, the functional roles of these mutant GPCRs in GI cancers remain unknown.

Besides mutations, GPCRs, like chemokine and histamine receptors (*HRH2*), exhibit significant copy number variations (CNVs) in tumors. Several broad-type GPCRs are universally overexpressed throughout the GI tract, regulating digestive and pathophysiological processes [54,63,64]. The upregulation of receptors like 5-HTRs, FFARs, HRs, PARs, EPs, and TGRs plays pivotal roles in proliferation, invasion, metastasis, and inflammation in the small intestine and colon. It has been reported that the CNVs of chemokine receptors, LPARs, and ARs contribute to the initiation and progression of hepatocellular carcinoma [65,66]. Early studies have implicated numerous viruses that harbored open reading frames and evolved to take advantage of the signaling network for replicative success by encoding GPCRs [67]. In GC, the Epstein-Barr virus (EBV/HHV-4) encodes a class A GPCR called BILF1, affecting multiple cellular pathways [68].

### 2.6. Widespread Mutations of G Proteins

As oncogenic drivers in multiple prevalent cancers, many G proteins are considered part of the cancer-associated gene panels routinely employed by a wide range of clinical oncology studies. MutSig2CV analysis indicates that *GNAS* is the most frequently mutated G protein in TCGA cohorts, concordant with the sequence results of the catalog of somatic mutations in the cancer (COSMIC) database. *GNAS* aberrations widely occur in tumors originating from the pituitary (28%), pancreas (12%), thyroid (5%), colon (6%), and a few other locations [69]. Previous studies revealed that the two most frequently mutated residents, Arg 201 [70,71] and Gln 227 [72], might be functionally important. The significance of these two sites has been first confirmed in pituitary tumors [73]. The disease-causing altered resident Arg 201 leads to the constitutive cAMP signaling by reducing the GTP hydrolysis of the active GTP-bound Gαs. However, the conclusion was reshaped by a recent structural study of *GNAS*, indicating that the stabilization of the intramolecular hydrogen bond network (H-bond network) plays a pivotal role in mutation-mediated constitutive activation [74]. These aberrations in *GNAS* are responsible for initiating and progressing multiple types of GI cancer, such as colon neoplasia, GC, and pancreatic adenocarcinomas (PDAs). In colon cancer, the *GNAS* can mediate the tumorigenesis of inflammatory factors by stimulating the Gs-Axin-β-Catenin pathway axis [75]. In the rare gastric adenocarcinoma, *GNAS* mutations were tightly associated with deep submucosal invasion and increased tumor size by activating the Wnt/β-catenin pathway [76]. Besides, at the early onset of invasive PDAs, frequent *GNAS* mutations (~41–75%) suppressed the PKA-mediated SIK and reprogrammed lipid metabolism in the precursor of PDAs [77,78].

Although *GNAQ* and *GNA11* mutations were less studied in tumors than *GNAS*, these mutations were well-established in Sturge-Weber syndrome [79] and leptomeningeal melanocytosis, arising from the central nervous system (~50%) [80], and also in the blue nevi and the primary uveal melanomas (UVM)/uveal melanoma metastases (83%) [81,82]. The somatic mutations are mainly located in the residues Q209 or R183, which are essential for GTP hydrolysis and cause constitutive activation due to loss of GTPase activity. In uveal melanoma (UVM), the more common Q209 mutations were more potent in tumorigenesis assays in nude mice models [82]. Consistently, the mutant GNAQ^Q209L^ contributed to MAPK pathway activation [81] and exhibited more significant activated ERK than the GNAQ^R183Q^ [83,84]. The activated GNAS mutant can also stimulate YAP-dependent transcription through a Trio-Rho/Rac signaling circuitry instead of the canonical Hippo pathway in UVM [85]. Furthermore, GNA13 is upregulated in several solid tumors, such as GC [86], nasopharyngeal cancer [87], breast cancer [88], squamous cell cancers [89], and colorectal cancer [90]. Interestingly, both GNA13 and RhoA have shown relevance to the transformation capacity and metastatic potential in epithelial cancer and fibroblasts, but the axis appears to play a tumor-suppressive role in B-cell lymphomas [91]. Large-scale sequencing of lymphoid and hematopoietic malignancies indicated that the mutant residues could be found throughout the gene [92,93].

The cDNA library screening distinguished the functionally relevant mutations of the Gβ proteins *GNB1* and *GNB2*. The gain-of-function alterations of these proteins can disrupt the interactions of Gα-Gβγ and constitutively stimulate the downstream signaling effectors, conferring resistance to targeted kinase inhibitors [94,95]. Recently, emerging variants in all five Gβ proteins have been reported, such as *GNB2* Arg52Leu in familial cardiac arrhythmia condition, Gly77Arg in neurodevelopmental disorder, and monoallelic missense variants in developmental delay/intellectual disability (DD/ID) [96]. Emerging evidence supports that Gβ mutants also occur in various cancer types and relate to distinct cancer subtypes. GPCRs also transduce the signal through β arrestins instead of G proteins, mediating GC cell invasion, migration, and epithelial-mesenchymal transition (EMT) [24]. For example, the protein kinase AKT exerts its oncogenic function through the signaling complex GPR39/β arrestin1/Src upon obestatin stimulation [97].

Some mutations occur in oncogenic kinase alterations, such as *BCR-ABL* fusion protein, *ETV6-ABL1*, JAK^V617F^, and BRAF^V600K^, to enhance the drug resistance of the corresponding kinase inhibitors [95,98]. Nevertheless, further investigations are needed into how these alterations influence tumorigenesis in different contexts. The potential roles of Gγ proteins, the close partners of Gβ proteins, should be clarified.

## 3. Aberrant GPCR Signaling in GC

GPCRs play hierarchical roles in many signaling networks. Dysregulations of the GPCRs extensively exist in tumor progression, metastasis, and immune response reprogramming. In recent years, aberrant GPCR members have been emerging in GC studies. This section will outline the updated findings of the GPCR signaling pathway in GC (Table 1).

### 3.1. Proliferation and Apoptosis

Mounting evidence has unveiled the multilayered crosstalk between GPCRs and proliferation- and apoptosis-related signaling circuits. The representative ones involve EGFR transactivation, MAPK cascades, the PI3K-AKT-mTOR pathway, and the Hippo signaling pathway [21,135].

#### 3.1.1. Transactivation in the EGFR and MAPK/ERK Pathway

GPCRs share many similarities with the tyrosine kinase receptors, such as EGFR and the MAPK/ERK signaling pathways [136], in regulating cell proliferation. The EGFR-mediated signaling pathway can be ligand-dependent or independent [137,138,139]. The “three membrane-passing signal (TMPS)” model is an EGFR ligand-dependent route. The activated RTKs are triggered by activated GPCRs and subsequently activate the extracellular signal-regulated kinase (ERK)/mitogen-activated protein kinase (MAPK) cascade. On the other hand, GPCR-mediated Src activation contributed to EGFR phosphorylation more directly. Both modes have been uncovered in GC. S1P could mediate the progression of GC via Gi- and matrix metalloprotease (MMP)-independent c-Met- and EGFR-transactivation [113]. However, the S1P- or LPA-induced transactivation of ERBB2 (also known as HER2) required the activation of MMP and the tyrosine kinase activity of EGFR [110]. In addition, the knockdown of the membrane-type bile acid receptor (M-BAR)/TGR5 suppressed the deoxycholate (DC)-induced phosphorylation of EGFR, and DC transactivates EGFR through M-BAR- and ADAM/HB-EGF-dependent mechanisms [124]. Infection with *H. pylori* boosted the expression of interleukin-8 (IL-8), which promoted cell proliferation by inducing EGFR transactivation [140]. Due to oncogenic activation and PGE_2_-EP4 pathway induction, the ubiquitous overexpression of the EGFR ligands and Adams have been identified in mouse gastric tumors [141]. PGE_2_-induced uPAR expression has also been implicated in the activation of Src, c-Jun NH_2_-terminal kinase (JNK), extracellular signal-regulated kinase (Erk), and p38 mitogen-activated protein kinase (p38 MAPK) [142,143]. GPCRs may also directly trigger MAPK cascades, establishing a connection between the external stimuli and their effect factors. These effectors may be further subdivided into four core categories: ERK1/2, JNK1-3, p38α-δ MAPKs, and ERK5. The LPAR2 inhibitor suppressed the proliferative and migration abilities of GC cell line SGC-7901 through the LPAR2/Gq11/p38 pathway, suggesting that LPAR2 might be a potential target for GC treatment [112]. Protease-activated receptor family (PAR1-4) also exerted pro-carcinogenic effects via the overactivated ERK1/2-MAPK pathway. For example, the reduction of EPCR impeded PAR1 activation, thus resulting in the downregulation of phosphorylated ERK1/2 and the suppression of the proliferation and migration of GC tumor cells [144].

#### 3.1.2. Activation of the PI3K-AKT-mTOR Pathway

PI3K is stimulated by the activated RTKs or GPCRs, ultimately leading to the synthesis of PIP3 and the recruitment of oncogenic effectors such as the serine/threonine kinase AKT. The PH domain in AKT permitted its binding with PIP3, contributing to the membrane accumulation and subsequent phosphorylation at T308 and S473 by PDK1 and mTORC2 [145]. Even though over 100 AKT substrates have been discovered in different settings, the associated mechanisms for most substrates have not been fully delineated [145]. mTOR is one of the AKT subtracts that is well-established to promote biosynthetic processes for cell growth. Since PI3K/AKT/mTOR signaling has also been identified as an ideal drug target for gastric carcinoma, the regulators may have a role in improving treatment design [146]. Indeed, some GPCRs have been proven to influence the activity of AKT in GC cells, such as the leucine-rich repeat-containing receptor Lgr6, adenosine receptor A2a, and the orphan receptor GPR39. Lgr6 was identified to empower GC cell proliferation by activating the PI3K/AKT/mTOR pathway [109]. Another GPCR, adenosine receptor A2a, was engaged in PI3K/AKT-regulated proliferation and migration in GC [123]. Additionally, GPR39 provided GC cells with a growth advantage by boosting the activity of AKT in an EGFR-dependent manner [97].

#### 3.1.3. Regulation of the Hippo Pathway

The canonical Hippo pathway kinase cascade is a critical tumor suppressor pathway, and its dysregulation has been widely implicated in organ size modulation and carcinogenesis [147]. The core components of the Hippo pathway are composed of STE20-like protein kinase 1/2 (MST1/2) and large tumor suppressor 1/2 and the major functional output Yes-associated protein 1 (YAP) and WW domain-containing transcription regulator protein 1 (WWTR1, also known as TAZ). Because there is a lacking DNA binding site in YAP/TAZ, TEF1-4 (TEAD1-4) is characterized as a bona fide transcription enhancer factor [147,148]. GPCRs have been found to control the Hippo pathway positively and negatively as a significant regulator of the intracellular pathway. The initial implication that GPCRs modulate Hippo signaling through LATS1/2 came from the study in serum starvation cells [149]. Two components, LPA and S1P, have been identified as the effective factors in serum that are responsible for YAP/TAZ activation through the recognition of the corresponding GPCRs. The LPA/S1P-mediated GPCR activation facilitates YAP/TAZ dephosphorylation via the G protein-cytoskeleton circuit. This study has laid the foundation for how YAP/TAZ senses the diffusible extracellular signals. However, several questions have also been raised after the initial discovery. Given that GPCRs constitute ~800 members and each GPCR can be coupled to diverse G proteins [19], the integrated effects on YAP/TAZ modulation remain elusive. The case can be more complicated when the dysregulation of G proteins and GPCRs is frequently determined in cancers [54]. Moreover, different G proteins stimulate the dephosphorylation of YAP/TAZ to various degrees. GPCRs can trigger YAP/TAZ activation by interacting with Gα_12/13_, Gα_i/o_, and Gα_q/11_ or suppress YAP/TAZ by binding with Gαs. However, which of these that Rho is involved with has not yet been identified; it is also unclear how the actin cytoskeleton regulates Lats1/2 phosphorylation. Emerging findings have revealed how specific GPCRs may fine-tune YAP/TAZ in given cellular surroundings [150]. The triggered LPA receptors have been demonstrated to play crucial roles in activating YAP/TAZ, causing tumor progression in the colon, ovarian, prostate, and breast [151,152]. The S1P-mediated S1P receptors contribute to hepatocellular carcinoma by coupling to Gα_12/13_ and stimulating YAP [153], connecting GPCR signaling to the Hippo pathway. Except for LPARs and S1PRs, the other GPCR-initiated signals can influence YAP/TAZ activity, including polypeptides (Angiotensin II, Thrombin, glucagon, etc.) [154,155], metabolites (purines, fatty acids, epinephrine, glutamate, etc.) [156,157], and hormonal factors (estrogen, endothelin-1, etc.) [158,159]. These signals have been widely indicated in human malignancies and are critical cell niche or microenvironment components. Recent studies pointed out that the mesenchymal niche manipulated the initiation of colorectal cancer by the rare peri-cryptal Ptgs2-expressing fibroblasts, and these fibroblasts exhibited paracrine control over tumor-initiating stem cells via the PGE2-EP4-Yap signaling axis [160]. The GPCR-Hippo crosstalk was also identified in GC stem-like cells: PAR1 stimulated the Hippo-YAP pathway and affected invasion, metastasis, and multidrug resistance [161]. As such, the GPCR regulation of YAP/TAZ has emerged as a driver, or as a potential therapeutic target, in gastric neoplasia. However, another study has found that AMOT, rather than Lats1/2, serves as the bridge between GPCR-mediated cytoskeleton changes and YAP/TAZ modulation in uveal melanoma cells, with an activated mutation at Arg183 and Gly209 in *GNAQ* (encoding for Gαq) and *GNA11* (encoding for Gα_11_), respectively [85]. As a result, the findings provide novel explanations for the alternations in actin dynamics induced by GPCR signals, which are somewhat different from previous studies. Therefore, this warrants exploring the interplay between AMOT, Lats1/2, and the actin cytoskeleton in GC, as the mechanic stress is context-dependent.

#### 3.1.4. GPCR-Signaling Integration and Crosstalk with Other Pathways in GC

Besides the above signaling circuits, other pathways have also been linked to GPCR-mediated oncogenicity in GC. These pathways involve the Notch pathway [162], hedgehog (Hh) signaling [163], and the Wnt/β-catenin pathway [164]. The Hh pathway is crucial for GC cell growth and cancer stem cell maintenance, and its activation has been highlighted in diffuse-type GC [165,166]. Smoothened (Smo, a member of class F) and Gpr161 (an orphan member) can function as positive and negative regulators in the Hh pathway, respectively [167]. The Wnt/β-catenin pathway is involved in tissue homeostasis and embryonic development. As Wnt (Wingless/Int1) stimulates the frizzled receptor (FZD, class F GPCRs), both G-protein independent and dependent signaling can be established [164,168].

### 3.2. GPCRs-Driven Metastasis of GC

Metastasis is how cancer cells establish ‘bench-heads’ in other organs or anatomical sites instead of the initial lesion, and it is responsible for more than 90% of cancer-related mortality [169] (Figure 3A). The most prevalent sites for GC metastasis are the liver, lung, bone, and lymph nodes [170]. Since Paget’s ‘seed and soil’ hypothesis laid the fundamental basis for metastasis, many investigators have contributed to a better understanding of the process. Several studies have identified the sequential multistep in GC metastasis: invasion into the surrounding tissue and the degradation of the basement membrane (BM), intravasation into the blood vessels or lymphatic systems, survival and translocation to distant tissues, extravasation into the foreign environment, and finally, colonization to proliferate and form a macroscopic secondary neoplasm [170,171,172]. As the complexity and relevance of metastasis have previously been widely reviewed, we will focus mainly on the roles and mechanisms of GPCRs during the invasion, BM degradation, and angiogenesis processes in GC.

#### 3.2.1. Inducing Epithelial-Mesenchymal Transition (EMT), Migration, and Invasion

Epithelial-to-mesenchymal transition (EMT) is regarded as the initial step for invasion, featuring a loss of cell polarity and integrity and the acquisition of motile mesenchymal characteristics. The pathologic activation of the EMT program is primarily executed by transcription factors (including SNAI1/2, TWIST1/2, and ZEB1/2) and microRNAs, ultimately resulting in the accumulation of the genes associated with mesenchymal properties, such as vimentin, fibronectin, and N-cadherin [173]. Though the above-mentioned molecular mechanisms are still lacking regarding GPCR-driven EMT in GC, GPCR signaling dysregulation is still frequently connected to EMT, migration, and invasion processes via dynamically regulating the downstream effectors and downstream cascades [174].

Although chemokines are tiny polypeptides (8-14kDa), they display pleiotropic effects in cancers. The chemokine system comprises nearly 50 chemokines that bind to 20 different chemokine receptors or four atypical chemokine receptors (ACKRs) [175]. This superfamily is distinguished by a substantial degree of redundancy, inferring that these chemokines can bind to different clusters of receptors and vice versa [176]. Intrinsic genetic or epigenetic regulators governed their expression and environmental cues such as hypoxia, microbiota, and metabolic [176]. For example, epigenetic regulator histone deacetylase 1 (HDAC1) suppressed CXCL8 expression by antagonizing the active nuclear transcription factor NF-κB [177]. Hypoxia has been revealed to induce the expression of CXCR4, CXCR7, and CXCL12 in different cancer cells, with the binding sites of hypoxia-inducible factor 1 (HIF1) as the promoters of these genes [178,179,180]. In GC, the elevated CXCL8 concentration was proved to be tightly correlated with the tumor stage instead of *H. pylori* infection, as shown in previous studies [181,182,183]. Several clinical investigations have suggested that the upregulation of chemokines and receptors was associated with GC pathogenesis, indicating that the specific chemokines might serve as potential diagnostic and therapeutic targets [184,185].

Chemokine receptors have attracted considerable attention due to their involvement in GC metastasis. A notable correlation was found between CCR7 expression and gastric carcinoma lymph node metastasis via stepwise regression analysis [186,187]. Strikingly, about 67% of primary gastric tumors exhibited CXCR4-positive expression [188]. The high concentration of the CXCR4 ligand CXCL12 has been validated in the malignant ascitic fluids from peritoneal carcinomatosis, and elevated CXCL12 is tightly correlated with the dissemination of GC cells to distant organs [188]. CXCL12-stimulated CXCR4 enhanced NF-κB and STAT3 signaling activation and, in turn, led to its transcriptional upregulation, which formed a positive feedback loop. This loop is linked to EMT, migration, and invasion in GC [189]. In response to CXCL12, CXCR4 also conferred the GC cell EMT and metastasis process via stimulating mTOR and some well-known oncogenic kinases: EGFR, SRC, or c-MET [190,191]. The crosstalk between TGF-β1 and the NF-κB pathway was triggered by the CCL2-CCR2 axis, leading to EMT-related protein upregulation [192]. Besides, chemokine receptors also induced organ-specific metastasis. CXCR4 and CCR7 are the primary receptors guiding the metastasizing cells, including GC cells [193]. Moreover, the high levels of CCR9 in melanoma, breast, and ovarian cancer make them efficiently translocate to the highly CCL25-expressing small intestine [194,195,196].

Many other GPCRs also govern the development of GC invasion. For example, GPER1 inhibition blocked EMT in GC cells by inhibiting the PI3K/AKT pathway [197]. Similar regulation that is mediated by adenosine receptor 2 (A_2a_R) or GPR30 could also be observed in GC [122,123]. In addition, the MAPK cascades were activated by the formyl peptide receptor 2 (FP_2_R), S1P_2_R, muscarinic acetylcholine receptor 3 (M3R), P2Y receptors (P2YR), and γ-Aminobutyric acid receptor A (GABA_A_), thus contributing to the invasion and metastasis in GC [104,105,114,119,121,128]. Many other GPCRs have also been linked to GC metastasis, while the underlying mechanisms are unknown. For instance, the angiotensin II receptor type 1/2 (AT1R/AT2R) has been locally upregulated and indicated to carry a much higher risk of nodal spread [101].

#### 3.2.2. Degrading the Barriers to Invasion

BM, a specialized extracellular matrix (ECM), plays a critical role in normal epithelium tissue architecture. BM disruption is a must for cancer cells leaving the primary location, controlled by the balance between the expression of MMPs and their tissue inhibitors (TIMPs) [198,199]. The expression of MMPs was upregulated by a histamine-H2 receptor or Thrombin-PAR1 signaling [143]. *H. pylori* was reported to be crucial during the invasion by upregulating cyclooxygenase-2 (COX-2) through ATF2/MAPK stimulation. The COX-2 inhibitor or EP2 receptor antagonist repressed angiogenesis and tumor invasion via the uPA system, which is a determinant factor in transforming the zymogen plasminogen into plasmin for degrading the ECM constituents [200]. Furthermore, the bacterial pathogen of *H. pylori’s* consistent infection manipulated a variety of extracellular proteases [201], but the exact mechanisms need further exploration. The interactions between microbial metabolites and GPCRs may provide new insights into the complicated process [202].

#### 3.2.3. Driving Angiogenesis

Angiogenesis is the process of vessel splitting from pre-existing vessels and is essential for tumorigenesis and progression, especially for those solid tumors exceeding 1–2 mm in diameter, as it provides oxygen and nutrients [203,204]. Many GPCRs exerted pro-tumor effects by promoting tumor-associated angiogenesis, notably Thrombin receptors, S1PRs, lysophosphatidic acid receptors (LPARs), and Prostaglandin receptors (Figure 3B). PAR1 is necessary for physio-pathological angiogenesis since poor vasculature development results in animal embryos dying after PAR1 deprivation. Thrombin-mediated PARs cleavage upregulates the transcription of many proangiogenic genes, such as VEGF and its receptor VEGFR, MMP2, angiopoietin-2 (Ang-2), and others [99,100,205,206]. Moreover, endothelial differentiation gene 1 (Edg1)/S1P_1_R is the first reported GPCR in blood vessel formation. Furthermore, the intrauterine death of Edg1 ablation mice happened mainly due to abnormal angiogenesis [207,208]. The Gα12/Gα13-coupled receptors LPA4 and LPA6 synergistically regulate endothelial Dll4 expression through YAP/TAZ activation, which mediates sprouting angiogenesis [209]. Moreover, *H. pylori*-induced VEGF upregulation was activated through p38 MAPK COX2-PEG_2_-EP2/4 signaling [210]. Other orphan receptors are also involved in tumor angiogenesis, such as KSHV-GPCR, GPR124, ELDT1, and GPER [211].

### 3.3. Remodeling the Tumor Microenvironment (TME) to Promote Immune Escape

TME acts as a unique niche populated by multiple cell types (including cancer cells, immune cells, and stromal cells), ECM, and diverse secreted factors (such as exosomes and microRNAs) [212,213]. The altered TME landscape is related to tumor progression, metastasis, and therapeutic responses [214]. Recently, the sophisticated TME infiltration pattern of GC (termed as TMEscore) has been defined based on the assessment of 22 immune cell types and cancer-associated fibroblasts (CAFs), which were correlated with genomic characteristics and pathologic features [212]. The biology and function of CAFs have emerged as an area of active investigation and have been reviewed elsewhere [215,216]. The compositions of infiltrated immune cells within TME varied greatly, and one of the most important mechanisms involved the chemokines and their receptors [176].

It is noteworthy that chemokines and chemokine receptors can be ubiquitously expressed in tumor cells, immune cells, and stromal cells [217]. Alternations in chemokines and their receptors shaped the TME immune cell constitution and remodeled the immune responses, some of which are hijacked by tumor cells to avoid immune surveillance and elimination [218]. The antitumor immune responses were driven by the recruiting immune cells, mainly including dendritic cells (DCs), CD8+ T cells, natural killer (NK) cells, and M1 macrophages. GC with a high CXCR3 expression level was shown to have increased DC and T cell infiltration. The CXCR3/CXCL4 or CXCR3/CXCL4L1 axis is necessary to recruit DCs as they elicit potent antitumor functions through substantially stimulating T cells and activating the related humoral response [219,220]. Similarly, CXCR3 also plays a vital role in CD8+ T cell infiltration that directly damages the tumor cells after being differentiated into cytotoxic CD8+ T cells [221,222]. In addition, NK cells represent professional killer cells, whose accumulation in the TME is the consequence of upregulated CXCL10 and CXCL12 signaling through CCR7 or CXCR3 [223,224].

On the other hand, chemokine signaling is also involved in the formation of immune-suppressive TME, where tumors evolve to escape recognition and clearance. This process has been largely linked to the infiltration of diverse protumor immune cell populations, such as regulatory T (Treg) cells, the M2 macrophages, monocytic myeloid-derived suppressor cells (M-MDSCs), and granulocytic (or PMN-) MDSCs [176,225]. CCL22, mainly produced by tumor cells (or macrophage-mediated), causes an abundance of Treg cells in TME via interacting with the receptor CCR4 on the surface of Treg. Another receptor, CCR10, in Treg cells also facilitated their migration in response to CCL28 [226,227]. Moreover, the nonpolarized macrophages (M0) originating from the recruited monocytes can be differentiated into two main subtypes, M1 and M2 macrophages, exhibiting extremely distinct functions toward cancers. These transitions depended on a large spectrum of chemokine signals. Active monocyte recruitment required tumor-derived chemokine releases, such as CCL2, CCL3, CCL4, CCL5, CCL20, and CCL18. Additionally, the blockade of the CCL2-CCR2 circuit led to M2 macrophage accumulation, whereas CCL11 skewed macrophages toward an M2 phenotype [228,229,230,231,232,233]. MDSCs were subdivided into two major groups: polymorphonuclear MDSCs (PMN-MDSCs) and mononuclear MDSCs (M-MDSCs). CXCR2 specifically mediated the migration of PMN-MDSCs to the tumor site by binding with CXCL1/CXCL2/CXCL5, whereas the accumulation of M-MDSCs requires CCL2-CCR2 interaction. Functionally, MDSCs employed diverse mechanisms to suppress T cell functioning, mainly through releasing high levels of arginase 1 (Arg1), reactive oxygen species (ROS), and nitric oxide (NO). Further research also suggested additional mechanisms, including the upregulation of COX2 and PGE2 in these MDSCs [234,235] (Figure 3C).

Other GPCRs are also involved in the regulations of immune responses. For example, prostaglandin (PG) production can mediate inflammation through its cognate GPCR EP1-EP4 (*PTGER1-4*). PGs, especially the PGE2, were produced by the cyclo-oxygenases COX-1and COX-2, the inhibitors (nonsteroidal anti-inflammatory drugs (NSAIDs)) of which have been utilized to comfort pains and reduce the incidence of a broad range of cancer types. The role of PGE2 has been extensively studied for inducing inflammation by stimulating other signaling pathways, including the Toll-like receptor (TLR)/MyD88 pathway [236], Wnt, and EGFR signal [237]. PAR1-deficient mice infected with *H. pylori* may suffer from severe gastritis due to lacking suppressing macrophage cytokine secretion and cellular infiltration [238]. In addition, TGR5 antagonized gastric inflammation by inhibiting the transcription activity of NF-κB signaling [239].

The involvement of GPCRs in immune remodeling is summarized in Table 2.

## 4. Therapeutic Strategies for Targeting GPCRs in GC

Despite the improving clinical outcomes, advanced GC patients benefit little from traditional surgery or chemotherapy and suffer from painful lives [240]. Personalized medicine and targeted therapy have been introduced to clinical applications for over two decades. For instance, trastuzumab has been integrated into the treatment for HER2-expressing patients, and ramucirumab has been utilized for VEGFR2-positive GC individuals [241]. Immune checkpoint inhibitors (ICI) also have been investigated as a frontline treatment [242,243,244,245]. Meanwhile, biomarkers and novel targeted therapies have been intensely investigated for advanced GC [246]. Substantial progress has been made by deciphering the functions of GPCR members in GC progression. However, only a handful of drugs that target GPCRs have been conducted in clinical trials for GC treatment (Table 3).

In order to accelerate the GPCR-targeted drug development for GC, many groups have identified potent compounds to inhibit or enhance the activity of GPCRs. However, the structures are only available for small partial GPCRs (~50 GPCRs), and 54% of GPCRs are orphans that are under-exploited. Machine learning approaches may be employed for predicting the interaction between immersed compounds and GPCRs based on established high-quality structural models [248]. With the evolving knowledge of GPCR pathways, we will be able to identify more effective drugs in formats, tissue-specific drug delivery systems, and appropriate treatment periods. Small molecules are the most prevalent GPCR modulators, while biologics are receiving more and more attention because of their versatility and specificity [249]. Antibodies, including antibody fragments and variable antibody domains, function with great penetration traits and are attracting considerable interest in drug development. Downregulated targeted GPCRs via RNA interference (RNAi) can represent potential approaches to gene therapy [250].

More efficient drug delivery systems with enhanced solubility and stability, lower dosages, and less toxicity have been developed, such as nanomaterials, nanocarriers, nanoconjugation, and nanoencapsulation techniques [251,252]. Furthermore, several solid tumors have well-established patient-derived xenografts (PDX) and xenograft-derived organoid models. These preclinical platforms recapitulated the genotypic and phenotypic landscape, endowed with a high predictive value for high-throughput drug screening. Nevertheless, they still have limitations, such as intratumor heterogeneity, compromised immune systems, and diverse tumor environments in GC [253].

## 5. Summary and Future Perspectives

GPCRs govern multiple signaling pathways and regulate GC development in various aspects. The heterogeneous and complicated characteristics of GPCRs contribute to GC heterogeneity and result in the current untimely diagnosis and inefficiency of therapeutic applications. Not only can GPCRs transduce the extracellular changes to the intracellular signaling circuits, but the conformational changes of GPCRs can also continuously influence intracellular events. Aberrant GPCR activation and mutated GPCRs/G proteins can fuel cancer cell proliferation, migration, invasion, angiogenesis, and metastasis. In addition, the dysregulation of GPCRs affords advantages for immunosuppressive TME and drug resistance to malignancies.

Although much progress has been made on novel biomarker identification and molecular mechanism investigation, the current GPCR-based diagnosis and therapy in GC are far from clinically available. It is urgent that GPCR signaling-targeted therapy be developed. In future studies, several issues need to be addressed. First, as the mutation rates of GPCRs and G proteins are prominent in some cancer types, the development of small molecules that target the driver mutations is urgent. Second, because of the heterogeneity of the cancer cells and tumor microenvironment, we need to comprehensively appraise the activation of GPCR signaling and its crosstalk by using cutting-edge techniques such as scRNA-seq or scDNA-seq. Last but not least, more preclinical models based on patient-derived samples, such as organoids or xenografts, need to be developed to evaluate the efficacies and side effects of the screened drugs. With the deep investigation of the molecular mechanisms of GPCR signaling and the multicenter clinical trials, more therapeutic strategies will be delivered for targeting GPCR signaling, which will benefit GC patients.

## Figures and Tables

**Figure 1 cancers-15-00736-f001:**
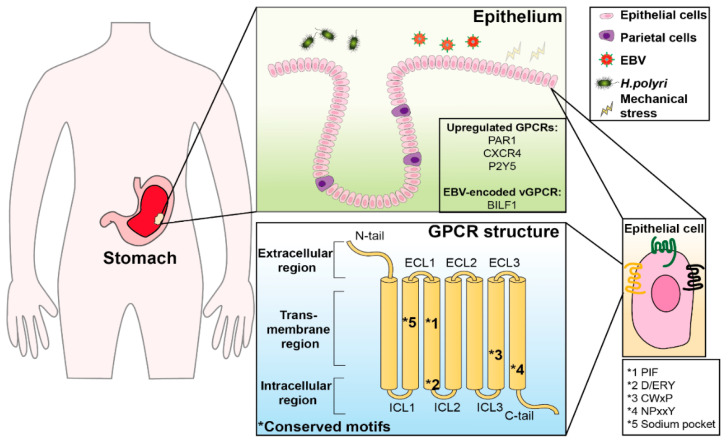
The structure GPCRs and their dysregulation in GC (*, Conserved motifs). GPCRs are widely expressed in the stomach. GPCRs participate in a variety of physiological and pathological processes. The upregulated GPCRs, including PAR1, CXCR4 and P2YR, and BILF1, are identified in GC. The five key sequence motifs in the class A GPCRs represent the most frequent mutant sites, which are conserved and responsible for their structural integrity and essential function. Abbreviations: GC, gastric cancer; PAR1, Protease-activated receptor 1; CXCR4, Chemokine CXC receptor 4; P2YR, P2Y receptor.

**Figure 2 cancers-15-00736-f002:**
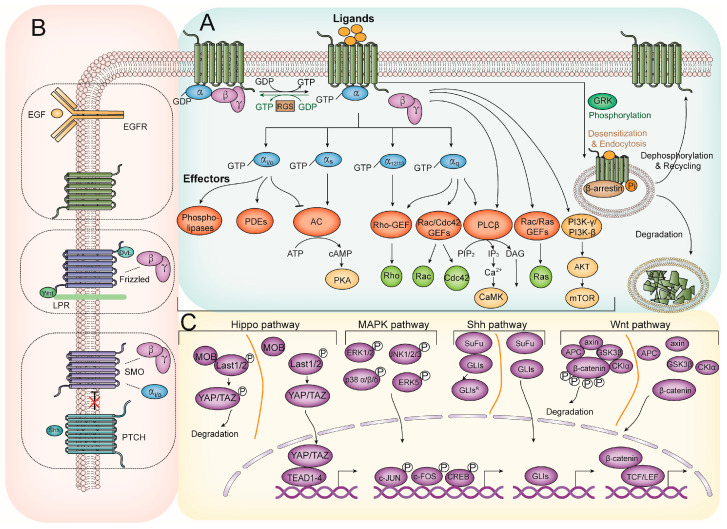
GPCR signaling and its crosstalk with other signaling pathways. (**A**) The most common GPCR-related signaling pathways. Agonist-stimulated GPCRs undergo a conformational change and facilitate the dissociation of Gα/Gβγ heterotrimer by replacing GDP with GTP on the Gα subunit. Subsequently, Ga and Gβγ trigger several downstream effectors, including secondary messenger systems, GEFs, Rho, and Ras GTPases, leading to a wide range of biological regulation. Besides the regulators of the G protein, signaling proteins (RGS proteins) promote the heteromeric complex reassociation and the signaling termination by accelerating intrinsic GTPase activity. Notably, agonist-activated GPCRs are also phosphorylated by GRKs and interact with β-arrestin, resulting in signaling desensitization and GPCR endocytosis. The endocytic β-arrestin-GPCR complex can be modulated by multiple factors and undergo degradation or recycling. (**B**) GPCR-associated crosstalk on the membrane and GPCR-EGFR crosstalk contain EGFR ligand-dependent transactivation and EGFR ligand-independent transactivation. The following pathways are the Wnt and Shh pathways. (**C**) The main pathways targeted by the multiple effectors in (a) consist of the following signaling pathways: Hippo pathway, MAPK pathway, Shh pathway, and Wnt pathway. Abbreviations: AC, adenylyl cyclase; AKT, protein kinase B; CREB, cAMP response element-binding protein; EGF, epidermal growth factor; EGFR, EGF receptor; ERK, extracellular signal-regulated kinase; GEF, guanine exchange factor; GLI, glioma-associated oncogenes; GPCR, G protein-coupled receptor; GRK, G protein-coupled receptor kinase; JNK, c-jun N-terminal kinase; LATS, large tumor suppressor kinase; MAPK, mitogen-activated protein kinase; mTOR, mammalian target of rapamycin; PDEs, phosphodiesterases; PI3K, phosphatidylinositol-3-kinase; PKA, Protein Kinase A; PLCβ, Phospholipase C β; ROCK, Rho-associated protein kinase; Shh, sonic hedgehog protein; SMO, Smoothened protein; SuFu, suppressor of fused; TAZ, transcriptional coactivator with PDZ-binding motif; TCF/LEF, T-cell factor/lymphoid enhancer factor; TEAD, transcriptional enhanced associate domain; YAP, yes-associated protein.

**Figure 3 cancers-15-00736-f003:**
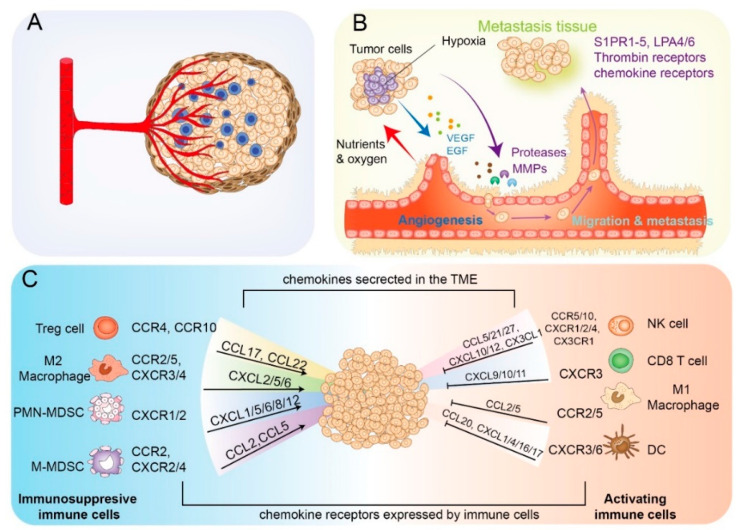
GPCR-mediated metastasis and tumor microenvironment remodeling in GC. (**A**) The TME of GC consists of blood vessels, lymph vessels, immune cells, stromal cells (including fibroblast, pericytes, and adipocytes), extracellular matrix (ECM), and secreted soluble factors, such as proteins, RNAs, and small organelles. (**B**) GPCRs control the process of angiogenesis and metastasis. GPCR activation drives the production of stimulatory angiogenic factors like VEGF and EGF. These factors promote the development of new blood vessels by modulating the mitogenesis, migration, and sprouting of endothelial cells (ECs). Moreover, several GPCRs regulate the metastasis process by influencing ECM, degrading the status of cancer cells (EMT, migration, and invasion), and colonizing foreign sites. (**C**) Chemokine–chemokine receptors modulate immune responses. The chemokines are secreted by tumor cells, immune cells, and stromal cells. The interaction of chemokine and specific chemokine receptors recruits antitumor immune cells and immunosuppressive immune cells into the tumor microenvironment.

**Table 1 cancers-15-00736-t001:** The most reported GPCRs in GC.

GPCRs	Ligand	Expression	Mechanisms	Biological Function	References
	Class A Receptors				
	Peptide/Protein Receptors				
Protease-activated receptors (PAR)	Proteases, such as Thrombin, TFLLRN (synthetic PAR1-targeted peptide)	PAR1/F2R: upregulation PAR2/F2RL1: upregulation PAR3/F2RL2: - PAR4/F2RL3: -	*H. pylori*→ERK/PI3K-AKT→α-arrestin→PAR1→CXCL2 PAR2→MAPK→VEGF/COX-2	Inflammation, angiogenesis	[99,100]
Angiotensin receptors (ATR)	Angiotensin II	AT_1_R: upregulation AT_2_R: upregulation	AT_1_R→VEGF	Angiogenesis, metastasis	[101,102]
Endothelin receptors (ETR)	Endothelin-1	ET_A_R: upregulation ET_B_R: - ET_C_R: -	ET_A_R→VEGF ET_A_R→β arrestin/Src→EGFR	proliferation, metastasis	[103]
Formyl peptide receptors (FPR)	fMLF, capthespin G	FP_1_R: - FP_2_R/ALX: upregulation FP_3_R: -	FP_1_R→ALOX5/15, SPMs (RvD1 and LXB4), SPM receptors (BLT1, ChemR23, GPR32) FP_2_R→MAPK	FP1R: inhibiting angiogenesis and proliferation FP2R: invasion and metastasis	[104,105,106]
Cholecystokinin receptors (CCKR)	CCK, gastrin	CCK1R: upregulation CCK2R/GR: upregulation	gastrin/GR→PKC→IκB, NF-κB	proliferation	[107]
Leucine-rich repeat-containing receptors (LGRs) group B	R-spondin1/2/3/4	Lgr4: upregulation Lgr5: upregulation Lgr6	Lgr4/5/6→β catenin Lgr6→PI3K/AKT/mTOR	proliferation, metastasis	[108,109]
	Lipid receptors				
Lysophosphatidic acid receptors (LPAR)	Lysophosphatidic acid	LPA1/Edg-2: - LPA2/Edg-4: upregulation LPA3/Edg-7: -	LPAR2→tyrosine phosphorylation of c-Met LPAR2→Gq11→p38	migration	[110,111,112]
Sphingosine-1 phosphate receptors (S1PR)	Lysophosphatidic acid: S1P	S1P_1_R/Edg-1 S1P_2_R/Edg-5 S1P_3_R/Edg-3 : ubiquitously expressed S1P_4_R/Edg-6 S1P_5_R/Edg-8	S1P_1_R→RAC-CDC42→ERK S1P→EGFR, c-Met	S1P_1_R & S1P_3_R: promote proliferation and migration, angiogenesis S1P_2_R: inhibit migration	[113,114]
Prostaglandin receptors (EPR)	PGE2	EP_1_R: - EP_2_R: upregulation EP_3_R: - EP_4_R: -	PGE2→DNMT3B→5mC enrichment (DNA hypermethylation) *H. pylori*→PGE2 upregulation→macrophage infiltration	proliferation, angiogenesis	[115,116]
	Chemokine receptors				
Chemokine CXC receptors (CXCR)	CXCL12-CXCR4/CXCR7CXCL8-CXCR1/CXCR2CXCL16-CXCR6	CXCR1: upregulated CXCR2: upregulated CXCR3 CXCR4: upregulation CXCR5 CXCR6: upregulation CXCR7	CXCL12/CXCR4→PI3K/Akt/mTOR CXCL12/CXCR4→ERK1/2 *H. pylori*→CXCL8→AKT/ERK/cyclin D1/EGFR/Bcl2/MMP9/MMP2	proliferation, migration, invasion, angiogenesis, metastasis	[117]
Chemokine CC receptors (CCR)	CCL2-CCR2CCL5-CCR5CCL19/CCL21-CCR7	CCR1/3/4/5/6/8/9: - CCR2/7: upregulation	CCR7→TGF-β1/NF-κB	migration, invasion, survival, metastasis	[118]
	Aminergic receptors				
Muscarinic acetylcholine receptor	Acetylcholine, carbachol,	M1R: M3R: upregulation M2R/M4R/M5R: -	M1R-TRPC6→PKC M2R/M4R→PKA→neurotransmitter release M3R→EGFR→MAPK/ERK M3R→Wnt pathway→YAP	proliferation, migration, invasion,	[119]
β-adrenergic receptor (β-AR)	isoproterenol	β1-adrenergic receptor: - β2-adrenergic receptor: upregulation β3-adrenergic receptor: -	ADRB2→NF-κB/AP-1/CREB/STAT3/ERK/JNK/MAPK→VEGF/MMP2/MMP7/MMP9	proliferation, invasion, metastasis	[120]
	Nucleotide receptors				
P2Y receptors (P2YR)	ATP	P2Y4: upregulation P2Y6: downregulation P2Y1/2/11-14: -	P2Y6→β catenin→ c-Myc P2Y2→Gaq→p38-MAPK/ERK/JNK	proliferation	[121]
Adenosine receptors (AR)	adenosine	A_1_/A_3_: - A_2a_R: upregulation A_2b_R: upregulation	A_2a_R→PI3K-AKT-mTOR A_2a_R→PKA/PKC	proliferation, metastasis	[122,123]
	Steroid receptors				
Membrane-type bile acid receptor (M-BAR/TGR5)	Deoxyolate, bile acids	TGR5: upregulation	TGR5→EGFR/MAPK	proliferation	[124]
	Orphan receptors				
GPR30	G1	GPR30: upregulation	GPR30→cAMP/Ca^2+^ GPR30→EGFR→PI3K/AKT/ERK	invasion, metastasis	[125]
GPR39	Obestatin	GPR39: -	GPR39→EGFR/MMP→AKT GPR39/β-arrestin/Src→EGFR→AKT	proliferation	[97]
	Class B receptors				
	Hormone receptors				
Growth hormone-releasing hormone (GHRH) receptor (GHRHR)	GHRH	GHRHR: upregulation	GHRHR→PAK1→STAT3/NF-κB	proliferation, inflammation	[126]
	Class C receptors				
	Ion receptors				
Calcium-sensing receptor (CaSR)	calcium ions	CaSR: upregulation	CaSR→Ca^2+^/TRPV4/β-Catenin	proliferation, migration, invasion	[127]
	Amino Acid receptors				
γ-Aminobutyric acid (GABA) receptor	GABA	GABA_A_: upregulation GABA_B_: -	GABA_A_→ERK1/2	proliferation, invasion	[128]
Metabotropic glutamate receptors (mGluRs)	Glutamate	mGluR5: upregulation mGluR1/5 (group I): - mGluR2/3 (group II): - mGluR4/6/7/8 (group III): -	mGluR5→ERK1/2	proliferation	[129]
	Adhesion receptors				
ADGRE5 (CD97)	CD55, α5β1 integrin, CD90	ADGRE5: upregulation	ADGRE5→MAPK	proliferation, metastasis	[130]
	Class F receptors				
Fizzled receptors	WNT, lipoglycoproteins	FZD2/6/7: upregulation FZD1/3/4/5/8/9/10: -	FZDs→Wingless/Int-1 (WNT)	proliferation	[131,132]
Smoothened receptors (SMO)	cholesterol, sterol	Smo: upregulation	SMO→HH	proliferation, invasion	[131,133]
	Viral receptors				
EBV-encoded vGPCR	metal ion (Zn^2+^)	BILF1: upregulation	BILF1→MHC class 1	proliferation, immune evasion	[68,134]

**Table 2 cancers-15-00736-t002:** Immune cell infiltration induced by chemokine and related receptors.

Cell Type	Receptors	Chemokines	Mechanisms Underlying Recruitment	Effects on Tumor Cells after Recruitment	References
Anti-tumoral immune cells					
Dendritic cell	CXCR3, CXCR6	CXCL4, CXCL1, CXCL16, CXCL17, CCL20	IFN-γ-induced chemokines production, *H. pylori* involvement	The most potent professional antigen-presenting cells, activation of cellular immunity, and T cell-dependent humoral immunity	[219,220]
CD8 T cell	CXCR3	CXCL9, CXCL10, CXCL11	CAFs-mediated IL6 secretion, tumor cell chemokines secretion, adhesion molecules (ICAM-1, VCAM-1)	Differentiated into cytotoxic CD8+ T cells to destroy tumor cells or memory CD8+ T cells to recirculate in the blood	[221,222]
NK cell	CXCR1, CXCR2, CXCR4, CX3CR1, CCR5, CCR10	CXCL10, CXCL12, CCL21, CX3CL1, CCL5, CCL27	Chemokine signaling regulated by HLA-G and CD47; stromal barriers	Cytokine production and cytotoxicity on tumor cells through STAT3; regulating DC maturation; modulating T cell activity	[223,224]
M1 macrophage	CCR2, CCR5	CCL2, CCL5	Disrupting NF-κB signaling or interacting with TNF-α;	High capacity to present antigens; proinflammatory cytokines (IL-1β, IL-1α, IL-12, TNF-α, and GFAP) production; stimulation of type-I T cell responses	[232,233]
Tumor-promoting immune cells					
Treg	CCR4, CCR10	CCL17, CCL22	Stimulation of JAK-STAT3 signaling pathway; remodeling of gastric microbiota by *H.pylori*.; stimulation of DCs due to *H.pylori.* infection	Suppressing CD4+ T cells, CD8+ cells, antigen-presenting cell (APC), monocytes, and macrophages; inhibitory cytokines like IL10, IL35, and TGF-β; inducing apoptosis by perforin/ granzyme production	[226,227]
M2 macrophage	CCR2, CCR5, CXCR3, CXCR4	CCL2, CCL5, CXCL9, CXCR12	STAT3 activation; PI3K/AKT/mTOR signaling pathway	Growth factors (FGF, VEGF, and IL-6) production; secreting matrix-degrading enzymes and cytokines	[230,231]
Monocytic MDSC	CCR2, CXCR2, CXCR4	CCL2, CXCL5, CXCL12	IL-6 production, JAK-STAT3 signaling,	High amounts of NO, Arg1, and immune-suppressive cytokines; suppression of nonspecific T cell responses	[234]
Granulocytic (or PMN-) MDSC	CXCR1, CXCR2	CXCL8, CXCL1, CXCL12, CXCL5, CXCL6	HGF/TGF-β/MCP-1 production, JAK-STAT3 signaling, IRF-8, NF-κB pathway, hypoxia	Large amounts of O^2−^, H_2_O_2_, and PNT (ROS) production; blocking T cell proliferation; depleting entry of CD8+ T cells to tumors	[234,235]

**Table 3 cancers-15-00736-t003:** Drugs and antibodies against GPCRs in GC clinical trials [6,135,247].

Drug Name	Targeted GPCRs	Types of Drugs	Tested Cancer Types	Status	NCT
Mogamulizumab	CCR4	mAb	Cataneous/Peripheral T-cell lymphoma; Adult T-cell lymphoma	Phase I: complete	NCT02946671
Vismodegib	SMO	small molecule	Basal-cell carcinoma; Head and neck cancer	Phase II: complete Phase II: complete Phase II: recruiting	NCT03052478 NCT00982592 NCT02465060
Sonidegib	SMO	small molecule	Basal-cell carcinoma	Phase I: recruiting Phase I: complete	NCT04007744 NCT01576666
Lutathera (Lutetium Lu 177 dotatate)	SSTR	peptide	Gastroenteropancreatic neuroendocrine tumors (GEP-NETs)	Most on recruiting	NCT04949282 NCT04727723 NCT04609592 NCT04524442 NCT02736500 NCT02489604 NCT04614766 NCT01860742
Lanreotide	SSTR	peptide	Advanced prostate cancer	Phase III: recruiting	NCT04852679 NCT03043664 NCT03017690 NCT02730104 NCT02736448

## Data Availability

Not applicable.

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
