# Peer review of "An Update of G-Protein-Coupled Receptor Signaling and Its Deregulation in Gastric Carcinogenesis"

_cancers, 2023, doi:10.3390/cancers15030736_

Round 1

Reviewer 1 Report

Yan et al, provide an important update on GPCRs in gastric cancers. This is a well thought out review and includes descriptions of all significant GPCR classes. They have the mechanisms and references right. The figures are also very nice and may lead to good citations.

Author Response

We sincerely thank you for the positive comments and appreciation of our work. Based on the other reviewers’ comments, we made some corrections to the manuscript to make the content more fluent and the description more accurate. 

Reviewer 2 Report

This review sounds like an interesting article. But some minor issues need to be fixed. The most crucial point is the lack of figure blocks. There are some schematic figures. However, the authors should add more figures related to the previously published papers and use them for the best understanding for the reader who reads this review. Moreover, please double-check the English language of the article by a native person. 

Author Response

Q1: This review sounds like an interesting article. But some minor issues need to be fixed. The most crucial point is the lack of figure blocks. There are some schematic figures. However, the authors should add more figures related to the previously published papers and use them for the best understanding for the reader who reads this review. Moreover, please double-check the English language of the article by a native person.

A1: Thank you for providing such suggestive comments. In the manuscript, we use three Figures to demonstrate the basic structure, the signaling crosstalks, and the promoting role of GPCR in gastric carcinogenesis. Apart from the Figures, we also included three integrated Tables to summarize the GPCR classification, immune cell-related GPCRs, and GPCR-targeted therapies, aiming to cover the most up-to-date research on GPCR in cancers, mainly gastric cancer. With these Figures and Tables, we believe this Review manuscript can deliver a clear and comprehensive message to readers.

Reviewer 3 Report

In the manuscript entitled “An update of G-protein-coupled receptor signaling and its deregulation in
gastric carcinogenesis”, Yan et al. examine the current knowledge about the role of GPCRs and heterotrimeric G proteins in gastric cancer (GC). Although the topic is thoroughly addressed the manuscript needs some major revisions to improve its comprehension and better define some issues concerning GPCRs and G proteins.
Major concerns:
1. Line 73 - “helix bundle end”: please, change to “C-terminus” since some GPCRs have no well- defined helical structure at the C-terminus.
2. Line 75 – “humans”: please, change to vertebrates.
3. Please, refer to “Alexander et al. The Concise Guide to PHARMACOLOGY 2019/20: G protein- coupled receptors. British Journal of Pharmacology (2019) 176, S21–S14” for GPCR classification.
4. Lines 107-108 – “GPCRs change their conformation and activate G-coupling proteins, such as β-arrestins and GPCR kinases (GRKs)”: please, change to “GPCRs change their conformation, activate G proteins, and couple with other proteins such as β-arrestins and GPCR kinases (GRKs)”.
5. Line 118 – “the Gα proteins”: please, change to “Gα subunits”.
6. Lines 126-127 – “The Gαi signaling activates phospholipase (PIs) and phosphodiesterases (PDEs), ultimately opening numerous ion channels”: please, change to “Members of the Gi family also activate phospholipase (PIs) and phosphodiesterases (PDEs), ultimately modulating
the opening of numerous ion channels”. In fact, the Gβγ complex dissociating from Gαi subunits can activate PIs while Gαt (the α subunit of Transducin, a member of Gi family, which is coupled with rhodopsin in retinal rod) activates PDE leading to cGMP decrease with closure of cGMP- dependent Na+/Ca2+ channels.
7. Lane 128: please, change “or” to “and”.
8. Line 129 – “1,4,5-triphosphate (PIP3) or diacylglycerol: please, change to “1,4,5-triphosphate  (PIP3) and diacylglycerol”. PLCs hydrolyzes PIP2 with the production of PIP3 and DAG, the later molecule activates PKC.
9. Line 130 – “conversion of PIP3 to PIP3”: do you mean PIP2 to PIP3?
10. Lines 180-181 – “It has been proposed that allosteric modulators can bind to an allosteric site instead of the orthosteric site for endogenous ligands”. It is well known in biochemistry and pharmacology that an allosteric ligand binds to a different site than the orthosteric binding site of a protein (receptor, enzyme, etc.). Therefore, the sentence isn’t needed.
11. Line 186-187 – please, remove the sentence since reference 29 reports that this PAM didn’t attenuate the side effects of a therapy, but it rather caused diarrhea.
12. Lines 257 and 259: metabotropic glutamate receptors are in class C while muscarinic receptors are in class A.
13. Other inaccuracies are present throughout the manuscript
14. The section in which the role of GPCRs in GC is delved deeper can be improved by exclusively reporting studies, which describe findings directly related to this cancer.
15. The English language must be improved. I got the feeling that some sections of the manuscript were better written than others.

Author Response

Response to Reviewer #3:

Q1: Line 73 - “helix bundle end”: please, change to “C-terminus” since some GPCRs have no well- defined helical structure at the C-terminus.

Response: Thanks for pointing out this issue, and we agree with your comments. The “C-terminus” is more accurate in the situation, and the “helix bundle end” has been replaced by “C-terminus”.

Q2: Line 75 – “humans”: please, change to vertebrates.

Response: The “humans” has been corrected to “vertebrates”.

Q3. Please, refer to “Alexander et al. The Concise Guide to PHARMACOLOGY 2019/20: G protein- coupled receptors. British Journal of Pharmacology (2019) 176, S21–S14” for GPCR classification.

Response: Thanks a lot for providing such a valuable reference on GPCR classification, and this paper has been cited in the revised manuscript (Reference #10).

Q4. Lines 107-108 – “GPCRs change their conformation and activate G-coupling proteins, such as β-arrestins and GPCR kinases (GRKs)”: please, change to “GPCRs change their conformation, activate G proteins, and couple with other proteins such as β-arrestins and GPCR kinases (GRKs)”.

Response: We appreciate these suggestions and we have revised the sentence as suggested.

Q5. Line 118 – “the Gα proteins”: please, change to “Gα subunits”.

Response: Thank you! We have revised the “the Gα proteins” to “Gα subunits”.

Q6. Lines 126-127 – “The Gαi signaling activates phospholipase (PIs) and phosphodiesterases (PDEs), ultimately opening numerous ion channels”: please, change to “Members of the Gi family also activate phospholipase (PIs) and phosphodiesterases (PDEs), ultimately modulating

the opening of numerous ion channels”. In fact, the Gβγ complex dissociating from Gαi subunits can activate PIs while Gαt (the α subunit of Transducin, a member of Gi family, which is coupled with rhodopsin in retinal rod) activates PDE leading to cGMP decrease with closure of cGMP- dependent Na+/Ca2+ channels.

Response: Thanks for the detailed explanation as a GPCR area expert. We thank the Reviewer for this detailed comment. The sentence indicated has been revised accordingly.

Q7. Lane 128: please, change “or” to “and”.

Response: We have changed “or” to “and”, as you suggested.

Q8. Line 129 – “1,4,5-triphosphate (PIP3) or diacylglycerol: please, change to “1,4,5-triphosphate  (PIP3) and diacylglycerol”. PLCs hydrolyzes PIP2 with the production of PIP3 and DAG, the later molecule activates PKC.

Response: Thanks for indicating this issue. We corrected “or” to “and” in the revised manuscript to describe the biochemistry process accurately.

Q9. Line 130 – “conversion of PIP3 to PIP3”: do you mean PIP2 to PIP3?

Response:  We apologize for this typo mistake. We have changed “PIP2” to “PIP3” in the updated revision.

Q10. Lines 180-181 – “It has been proposed that allosteric modulators can bind to an allosteric site instead of the orthosteric site for endogenous ligands”. It is well known in biochemistry and pharmacology that an allosteric ligand binds to a different site than the orthosteric binding site of a protein (receptor, enzyme, etc.). Therefore, the sentence isn’t needed.

Response: Thank you for the valuable suggestion. We have deleted the sentence.

Q11. Line 186-187 – please, remove the sentence since reference 29 reports that this PAM didn’t attenuate the side effects of a therapy, but it rather caused diarrhea.

Response: Thank you for the insightful comment. We have checked the additional reference paper and corrected this point in the manuscript.

Q12. Lines 257 and 259: metabotropic glutamate receptors are in class C while muscarinic receptors are in class A.

A12: Thanks for indicating the message. We added the information in the corresponding sentence.

Q13. Other inaccuracies are present throughout the manuscript

A13: Thanks again for the detailed check on our manuscript, and we have learned a lot from your rigorous attitude. We have re-checked all the parts and amended the inaccurate information in the manuscript.

Q14. The section in which the role of GPCRs in GC is delved deeper can be improved by exclusively reporting studies, which describe findings directly related to this cancer.

A14: Thank you for the suggestions. GPCR is well-studied in other GI cancer, while the studies in GC is relatively limited. We went through nearly all the findings and literature and tried to incorporate the current knowledge on how GPCRs contribute to GC pathogenesis. Hopefully, Table 1 and Figure 3 can illustrate the information better.

Q15. The English language must be improved. I got the feeling that some sections of the manuscript were better written than others.

A15: Finally, we thank this Reviewer for the insightful comments and suggestions. We invited a colleague with overseas working experience to review and revise the manuscript thoroughly. We hope the description have been greatly improved.

Round 2

Reviewer 3 Report

The manuscript was improved according to suggestions.